# Nivalenol Mycotoxin Concerns in Foods: An Overview on Occurrence, Impact on Human and Animal Health and Its Detection and Management Strategies

**DOI:** 10.3390/toxins14080527

**Published:** 2022-07-31

**Authors:** Pradeep Kumar, Dipendra Kumar Mahato, Akansha Gupta, Surabhi Pandey, Veena Paul, Vivek Saurabh, Arun Kumar Pandey, Raman Selvakumar, Sreejani Barua, Mandira Kapri, Manoj Kumar, Charanjit Kaur, Abhishek Dutt Tripathi, Shirani Gamlath, Madhu Kamle, Theodoros Varzakas, Sofia Agriopoulou

**Affiliations:** 1Applied Microbiology Laboratory, Department of Forestry, North Eastern Regional Institute of Science and Technology, Nirjuli 791109, India; madhu.kamle18@gmail.com; 2Department of Botany, University of Lucknow, Lucknow 226007, India; 3CASS Food Research Centre, School of Exercise and Nutrition Sciences, Deakin University, Burwood, VIC 3125, Australia; kumar.dipendra2@gmail.com (D.K.M.); shirani.gamlath@deakin.edu.au (S.G.); 4Department of Dairy Science and Food Technology, Institute of Agricultural Sciences, Banaras Hindu University, Varanasi 221005, India; salonigupta.ag@gmail.com (A.G.); surabhi@bhu.ac.in (S.P.); veena.paul20@gmail.com (V.P.); abhi_itbhu80@rediffmail.com (A.D.T.); 5Division of Food Science and Postharvest Technology, ICAR—Indian Agricultural Research Institute, New Delhi 110012, India; vivek.bhu12@gmail.com (V.S.); charanjitkaur6@gmail.com (C.K.); 6Food Science and Technology, MMICT & BM(HM) Maharishi Markandeshwar (Deemed to be University), Mullana, Ambala 133207, India; akpandey.fst@gmail.com; 7Centre for Protected Cultivation Technology, ICAR-Indian Agricultural Research Institute, Pusa Campus, New Delhi 110012, India; selvakumarsingai@gmail.com; 8Department of Agricultural and Food Engineering, Indian Institute of Technology, Kharagpur 721302, India; sreejani301@gmail.com; 9Centre for Rural Development and Technology (CRDT), Indian Institute of Technology Delhi (IITD), New Delhi 110016, India; kaprimandira@gmail.com; 10Chemical and Biochemical Processing Division, ICAR-Central Institute for Research on Cotton Technology, Mumbai 400019, India; manoj.kumar13@icar.gov.in; 11Department of Food Science and Technology, University of the Peloponnese, Antikalamos, 24100 Kalamata, Greece; t.varzakas@uop.gr

**Keywords:** nivalenol, food contamination, human and animal health, detection and management strategies

## Abstract

Mycotoxins are secondary metabolites produced by fungi that infect a wide range of foods worldwide. Nivalenol (NIV), a type B trichothecene produced by numerous *Fusarium* species, has the ability to infect a variety of foods both in the field and during post-harvest handling and management. NIV is frequently found in cereal and cereal-based goods, and its strong cytotoxicity poses major concerns for both human and animal health. To address these issues, this review briefly overviews the sources, occurrence, chemistry and biosynthesis of NIV. Additionally, a brief overview of several sophisticated detection and management techniques is included, along with the implications of processing and environmental factors on the formation of NIV. This review’s main goal is to offer trustworthy and current information on NIV as a mycotoxin concern in foods, with potential mitigation measures to assure food safety and security.

## 1. Introduction

Mycotoxins, which are secondary metabolites produced by many fungi, can be detrimental to both human and animal’s health when ingested through contaminated food [1,2,3,4,5,6]. The contamination of crops and their processed products is a major public health and economic concern. More than 80% of the agricultural produce is contaminated by at least one mycotoxin. Several genera of fungi, including *Cephalosporium*, *Cyclindrocarpon*, *Myrothecium*, *Phomopsis*, *Stachybotrys*, *Trichoderma*, *Trichotecium*, and *Verticimonosporium* species, are mainly responsible for the mycotoxin production among which the trichothecene group of mycotoxins share a major proportion, and hence pose wider concerns [7]. The trichothecene group of mycotoxins are categorized as type A, B, C, and D, which includes more than 170 different toxins, but their toxicity potential depends upon the type of mycotoxin being produced by the fungi [8].

Nivalenol (NIV) is one of the major mycotoxins, produced by several *Fusarium* species and belongs to the type B trichothecene. It is a natural contaminant found worldwide in various foodstuffs. Due to the potent cytotoxicity, NIV constitutes a serious health risk to both humans and animals. Its contamination is observed very frequently in agricultural products. Crops, such as wheat, barley, and corn, are usually contaminated with NIV, as these crops are more prone to the growth of NIV-producing fungal species [9,10,11]. Furthermore, NIV is also resistant against food processing conditions, thereby leading to the further spread of NIV-induced mycotoxicosis in consumers [12]. Controlling NIV exposure is especially important because of how frequently it contaminates foods and how easily it can harm mammals through immunotoxicity and hemotoxicity [13]. NIV can produce certain complications, such as inhibition of cell proliferation, induction of CXCL8/interleukin (IL) -8 secretions, and the involvement of stress-activated mitogen-activated protein kinases [14,15,16].

Chemically, the structure of NIV is very similar to DON (deoxynivalenol) mycotoxin of the type B trichothecene group. The only difference between NIV and DON is one oxygen atom at the C-4 position (Figure 1); however, the toxicity of these oxides of carbon completely differs from each other. The co-occurrence of NIV and DON has been reported extensively, but the oxidative stress and toxicity induced by NIV contamination are higher than that described for DON [7,12]. Several studies have reported the contamination of NIV in foods, including wheat [17,18], barley [19,20], groundnuts [21], maize and their by-products [22,23,24], but the level of contamination varies with the type of crop and ecological factors, such as temperature and humidity. Magan et al. [25] reported that environmental factors, including farm practices, handling, storage, processing, and distribution of food grains, largely influence the spread of NIV contamination. Therefore, the knowledge of climatic conditions on the growth of *Fusarium* species, along with its behavior and mycotoxin production, is essential to control the risk of NIV contamination in crops [26].

## 2. Chemistry and Biosynthesis of Nivalenol

Trichothecenes are sesquiterpenoids with a variable pattern of oxygen and ester groups on a core tricyclic skeleton containing an epoxide function, also known as the 12,13-epoxytrichothec-9-ene (EPT) [27]. A large variety of trichothecene have been found in diverse fungal taxa and are grouped into four classes (types A-D) based on the substitution pattern on the fundamental EPT core [28,29]. Numerous research has been conducted to understand the various impacts of trichothecenes and to study the underlying mechanisms to mitigate the huge losses caused by *Fusarium* diseases globally [30,31,32,33], which endanger the viability of farms and entire rural communities.

*Fusarium’s* two primary chemotypes are connected to organisms that either generate deoxynivalenol or nivalenol [34]. The functioning of two genes, TRI13 that expresses a trichothecene 4-hydroxylase and TRI7 that encodes a trichothecene 4-acetyltransferase, has been shown to be the genetic difference between deoxynivalenol and nivalenol chemotypes. Both genes are present but inactive in *Fusarium* chemotypes that produce deoxynivalenol [35,36]. Compared to its type B companion, DON and NIV, which are more abundant in Asia, contaminate grains at greater levels and cause more severe acute and chronic toxicities. In *F. graminearum*, twelve trichothecene genes are grouped within a 25-kb area on chromosome II that is inherited as a single core TRI cluster [37,38]. Two more loci are involved, including the single gene Tri101 and the two-gene Tri1-Tri16 locus, both of which are situated in unrelated chromosomal areas [39]. In *F. graminearum*, the presence of both functioning Tri7 and Tri13 genes is needed for NIV-producing chemotypes [35,40].

Biosynthetic clusters are used in fungi to produce secondary or specialized metabolites [41]. As a result, the proteins that catalyze the biosynthesis of distinct trichothecenes are likewise arranged in biosynthetic clusters (TRI) [37,38], where many of the genes appear to be coordinately expressed and controlled. Intermediates from two separate trichothecene routes, deoxynivalenol biosynthesis and T-2 toxin generation, can feed the nivalenol process. The entry processes from the deoxynivalenol biosynthesis into the nivalenol biosynthesis are mediated by TRI13, which codes for a trichothecene 4-hydroxylase, and TRI7, which codes for a trichothecene 4-acetyltransferase and is non-functional in *Fusarium* species of the deoxynivalenol chemotype.

The gene TRI13 encodes a trichothecene 4-hydroxylase that has been identified and described from *Fusarium graminearum* strain 88-1, which represents the nivalenol chemotype. This enzyme catalyzes the conversion of 3,15-acetyldeoxynivalenol to 3,15-diacetylnivalenol, hence functioning as a switch between deoxynivalenol-producing and nivalenol-producing *Fusarium graminearum* strains [35,42]. The TRI7-encoded trichothecene C-4-acetyltransferase then performs C-4 acetylation, leading to the formation of 3,4,15-triacetylnivalenol [35,42,43]. The same intermediate can also be produced via the trichothecene 7,8-dihydroxylase (TRI1), which acts as 3,4,15-triacetoxyscirpenol 7,8-dihydroxylase and converts 3,4,15-triacetoxyscirpenol, an intermediate of the T-2 toxin biosynthesis, into 7,8-dihydroxy-3,4,15-triacetoxyscirpenol, followed by a currently unknown protein that confers the carbonyl function to the C-8 of 3,4,15-triacetylnivalenol [44,45]. In *Fusarium graminearum* strain 79A1 [46], the trichothecene C-3 esterase (TRI8) deacylates 3,4,15-triacetylnivalenol, generating 4,15-diacetylnivalenol, which is subsequently deacylated at C-4 by an unknown protein. Furthermore, in the NIV chemotype, Tri13 hydroxylates 3,15- diacetyldeoxynivalenol to yield 3,15-diacetylnivalenol, which is further transformed into NIV [47].

## 3. Effects of Environmental Factors on Nivalenol Production

Environmental factors, including climatic conditions and farming practices, as well as handling and storage, processing, and transportation of food, all play a role in NIV contamination [25]. According to Nazari et al. [48], the production of NIV in durum wheat is initiated at the optimum temperature of 25 and 35 °C, when inoculated with *F. poae*. However, the fungus was unable to produce NIV at temperatures ≤ 10 °C [48]. The researchers also reported that the optimum temperature for NIV production by *F. poae* was greater than that of the optimum temperature for colony development (24.7 °C), implying that NIV is generated when the fungus is under stress, such as high temperatures. In accordance with these findings, Shen et al. [26] revealed that environmental conditions are one of the most major factors that determine the geographic distribution of *Fusarium spp.* and trichothecene chemotype variability in China, with a considerably higher predominance of *F. asiaticum* NIV chemotype in the southern, relatively warm regions of the country. Contrary to the above, Schöneberg et al. [49] observed that *F. poae*-inoculated cereals contain higher NIV at 10 °C, than those inoculated at 15–20 °C.

Water activity (a_w_) plays a vital role in the growth of a microorganism, and its toxin productions. Hope and Magan [50] reported that the highest NIV production was obtained between 0.99 a_w_ and 0.96a_w_, with the best result at 0.981 a_w_ with a 40 day incubation period for *Fusarium culmorum*. However, at 0.995a_w_, NIV production was found to be 10 times lower. In addition, Llorens et al. [51] reported that the *Fusarium* species showed the highest production of NIV of 161 μg/g at 20 °C and 0.970 a_w_. Moreover, no NIV production was detected at 32 °C or above. According to Neme and Mohammed [52], a_w_ below 0.9 ceases the production of NIV in *Fusarium* species. Apart from temperature and a_w_, substrate plays a vital role in NIV production by several microorganisms. Nazari et al. [48] reported wholemeal wheat flour to be the favourable substrate for NIV production under in vitro conditions. Vogelgsang et al. [53] reported maize kernels, followed by wheat, oat, and rice, to be a favourable substrate for NIV production. Moreover, Pettersson [54] reported barley to be the limiting cereal for NIV production.

## 4. Occurrence of Nivalenol in Food and Feed 

Several species of *Fusarium,* including *F. poae*, *F. culmorum*, *F. grainearum*, *F. cortaderae*, *F. asiaticum*, *F. nivale*, *F crookwellense* etc., are mainly responsible for producing NIV and its acetyle derivatives in foods [18,24,55,56]. The *Fusarium* species easily grow on cereal grain and produce *Fusarium* head blight disease under favorable environmental conditions. In general, moderate temperature and high air humidity are ideal for the infection and growth of these *Fusarium* species. However, sometimes it may vary from species to species, such as *F. graminearum* are usually found in the warmer and more humid regions (Australia, Eastern Europe, North America, South China), whereas cold climatic regions (Western Europe) are suitable for the growth of *F. culmorum* species [57]. NIV is reported in various cereals and cereal-based products worldwide (Table 1). It is also interesting to note that NIV mostly co-occurs with DON and is detected simultaneously in different food matrices. Furthermore, most of the methods mentioned in Table 1 were used for the multi-mycotoxin analysis.

## 5. Nivalenol Tolerance Limit

NIV has been noted in the past few decades to be present in wheat, barley, groundnuts, and maize at concentrations ranging from a few μg/kg to over mg/kg [13]. The highest contamination levels were found in cereals and cereal-based goods (107.2 µg/kg), followed by legumes and legume-based products (31.7 µg/kg). In cereal-based baby feeds, contamination levels were lower (17.1 µg/kg), whereas NIV was not discovered in alcoholic beverages [13]. There is currently a dearth of knowledge regarding the prevalence of NIV and its toxicological significance. In animal models, the long-term effects of NIV treatment at low oral dose levels have been investigated. While China places no NIV limit on meals and feeds, several nations only allow very low levels of NIV in grains [103]. The maximum limit for different mycotoxins has been established for various food commodities to protect human and animal health from agricultural produce that has been contaminated. To control the maximum level of NIV incidence in food and feed, the EU has not yet created any particular guidelines. Based on the scientific consensus regarding the general toxicity of NIV, the Scientific Committee on Food (SCF) set a temporary tolerated daily intake (t-TDI), or 0–0.7 µg/kg body weight per day, for NIV. However, 0.4 µg/kg body weight per day t-TDI for NIV has been set as the concentration level by the Japanese Food Safety Commission (FSCJ). Based on 10 years of data on the prevalence of NIV in 18 European countries, the European Food Safety Authority recently revised the TDI for NIV and set a TDI of up to 1.2 µg/kg body weight per day [104].

## 6. Mechanism of Toxicity and Health Effects of Nivalenol

The effects of NIV on cell viability have been investigated using a variety of cell lines and tests. The porcine jejuna epithelial (IPEC-J2) cells and human colorectal adenocarcinoma (Caco-2) cells are the most frequently used cell models in NIV cytotoxicity research because trichothecenes have been shown to alter cell proliferation, particularly in tissue with increased levels of cellular proliferation, such as intestinal epithelial cells. After 48 hours of exposure to NIV, Wan et al. [105] and Alassane-Kpembi et al. [106] used the 3-(4,5-dimethylthiazol-2-yl)-2,5-diphenyltetrazolium bromide (MTT) test to obtain comparable IC50 values on IPEC-J2 and Caco-2 cells, respectively, after 48 hours of exposure to NIV. More proof of the cytotoxicity of NIV after 48 hours of exposure is available (THP-1) on human monocytic leukemia cells and calf small intestinal epithelial cells B (CIEB) [23]. The effects of 24 hours of treatment, however, resulted in very different IC50 values in rat non-tumorigenic intestinal epithelial cells (IEC-6) and human hepatocellular carcinoma (HepG2) cells, Caco-2 cell clone C2BBe1 cells, and HepG2 cells, with the latter obtaining a higher IC50 value [107,108]. These results suggest that cells of human origin are more responsive to NIV after 24 hours than cells of other origins, in line with Taranu et al.’s [109] suggestion that human lymphocytes were more receptive to NIV than pig cells.

Most food commodities include several mycotoxins, as various fungal species and mycotoxins can co-occur in a single food product [1,110,111,112]. Multi-exposure to mycotoxin can have cumulative, antagonistic, or synergistic consequences. Furthermore, the toxicity of the mixture cannot always be predicted based only on the individual toxicity of each mycotoxin present. To provide more accurate and effective risk evaluations, research of the commonly reported toxin combinations is required. On the Caco-2/human colon cancer (HT29-MTX) 70/30 co-culture, the binary (NIV + fumonisin B1 (FB1), tertiary (NIV + DON + ZEA, NIV + DON + FB1, NIV + ZEA + FB1), and quaternary (NIV + DON + ZEA + FB1) combinations caused stronger cytotoxic effects on cell growth compared to NIV alone [113]. Contrary to this, on IPC-J2 cells, no increases in overall cytotoxicity of the binary, tertiary, and quaternary combinations of NIV with DON, ZEA, and FB1 were detected, as compared to the individual mycotoxin NIV. It has been shown that the NIV has an impact on the maturation of lipopolysaccharide-stimulated dendritic cells generated from murine bone marrow. It was demonstrated by Luongo et al. [114] that in vitro pre-treatment of DCs with NIV 3 mg/kg resulted in a reduction in the synthesis of MHC class II molecules. In humans, a growing body of research suggests that trichothecenes are frequently linked to gastroenteritis outbreaks [115]. Because the intestine is the first line of defense against food pollutants, such as mycotoxins, the intestine and intestinal mucosa might be exposed to high amounts of them after ingestion of contaminated food or feed. Cheat et al. [116] discovered significant gut mucosal alterations after acute NIV (1–10 mg/kg) exposure.

The histological scores of the pig jejunal explant cultures dropped in a concentration-dependent manner after 4 hours of exposure to NIV at doses of 1 mg/kg. NIV showed the highest amount of toxicity in comparison to the acute effects brought on by DON [116]. However, Alassane-Kpembi et al. [117] reported the synergistic effect of mycotoxins on the intestinal epithelium. They found that the simultaneous presence of mycotoxins in foods could be more toxic than predicted from the mycotoxins alone. NIV exposure generated strong pro-oxidant effects in IEC-6 cells, which were again larger than those caused by DON [118]. NIV (2 mg/kg) was found to cause mucin genes MUC5AC and MUC5B mRNA expression to be downregulated on Caco-2/HT29-MTX co-cultures, a well-known standard model for intestinal absorption, transport, and permeability studies. NIV is also suspected to contribute to the onset of Kashin–Beck Disease (KBD), an endemic, age-related degenerative osteoarthropathy that mostly affects adolescents on the Chinese mainland. Additionally, the induction of the pro-apoptotic protein Bax, the suppression of the anti-apoptotic protein Bcl-2, and the induction of caspase-3 activation in IEC-6 cells resulted in the formation of hypodiploid nuclei and a change in the distribution of the cell cycle, which in turn caused apoptosis, as shown by the induction of the pro-apoptotic protein Bax and the suppression of the anti-apoptotic protein Bcl-2 [107]. However, in dendritic cells, NIV (1–3 mg/kg) did not activate caspase-3, suggesting that the observed concentration-dependent decrease in expression levels was primarily brought about by necrosis rather than apoptosis induction. Smith et al. [119] observed that 12 hours of exposure to 0.8 mg/kg NIV and 0.006 mg/kg T-2 toxin caused cell necrosis primarily in human monocytic leukemia cells.

NIV inhibitors that bind to ribosomes can quickly activate mitogen-activated protein kinases (MAPKs) and cause death during the “ribotoxic stress response”. A crucial subject is the nature of the connection between ribosomal RNA damage and the emergence of MAPK signaling cascades. Unknown intermediary signal transduction pathways, likely to be protein-mediated, may take place between toxicant-damaged 28S rRNA and the MAP kinases [120]. This action may be limited to specific stages of the ribosome cycle. Although a variety of proteins interact with ribosomes and might conceivably modulate MAPK activation, no obvious mechanism has yet been discovered. Two kinases have been identified as potential mediators of the ribotoxic stress response generated by trichothecene-induced MAPK activation. These are protein kinases that are activated by double-stranded RNA (dsRNA) (PKR) [121]. PKR is a serine/threonine protein kinase that can be triggered by dsRNA, interferon, and other substances [122]. Hck belongs to the Src nonreceptor tyrosine kinases, a subfamily of protein tyrosine kinases known as “rheostats for immunological signalling” [123]. Figure 2 depicts an overview of the probable signaling cascades altered by NIV and their downstream consequences. However, further in-depth research is required to understand how these kinases transmit signals from the ribosome to MAPK cascades, eventually impacting gene transcription and mRNA stability.

## 7. Effects of Processing on Nivalenol 

Various processing methods, including physical processing (sorting, cleaning, milling, polishing) [124,125], thermal processing (cooking, baking, extrusion, per-boiling, roasting) [126,127], chemical treatments and fermentation processes [128] can affect NIV. According to Lešnik et al. [129], cleaning significantly affected the NIV content of the flour, where industrial cleaners showed an effective response with reduced NIV contamination in grains by 34.8%, whereas the traditional grain cleaner showed a decrease in NIV contamination by 15.5%. Schaarschmidt and Fauhl-Hassek [130] further reported more than 50% reduction in NIV with manual separation of unwanted kernels (such as split and poor-quality kernels, cob pieces). After manual sorting, industrial dry cleaning, floatation and washing further helped in the reduction in NIV contamination. Tibola et al. [131] also reported that preliminary processing, such as cleaning, sorting, and grading of wheat, significantly reduced the NIV content in grains. In addition, NIV dissolves in water, so cleaning of wheat grain followed by milling treatment effectively degraded 74% of NIV in two different wheat cultivars viz., *norin 61* and *chikugoizumi* [132]. Similarly, Thammawong et al. [133] reported decreased NIV content in milled flour compared to the contaminated whole grains. Lešnik et al. [129] reported that the level of contamination also depends on the type of milling and fractionation of grains. They revealed that increased NIV contaminations were observed in the outer kernel (bran and reduced flour) and reduced NIV contamination was observed in the endosperm fraction flour. The study also concluded that contaminated grains containing 130–200 μg/kg NIV showed reduced contamination up to 64.8–71% in roller-grinder mill, 38.1–59.9% in hammer crasher mill and 37–54.4% in millstone milling. Trigo-Stockli [134] showed a 20–69% reduction in NIV during dry milling of wheat flour.

Furthermore, Hossen et al. [135] showed that cooking of NIV contaminated noodles made from Japanese soft wheat cultivar namely *Chikugoizumi* decreased the NIV content to half. However, Sugita-Konishi et al. [136] reported a limited effect of heat/thermal treatment on the NIV content due to its thermo-stability. In addition, baking of barley flour at 200 °C for 70 mins reduced NIV by 77% [137]. In addition, Lešnik et al. [129] reported a 50% reduction in NIV in wheat bread during industrial and traditional baking processes. Furthermore, Bretz et al. [138] observed that NIV under heating in mild alkaline condition breaks into four different compounds, namely norNIV A, norNIV B, norNIV C, and NIV lactone.

In addition, Scudamore et al. [139] reported that extrusion of wheat flour at 140 °C with low moisture results in degradation of NIV. However, it was clear from the study that increased temperature during the extrusion process did not affect the NIV concentration, whereas the moisture content during extrusion was found to play a significant role in lowering NIV contamination. Furthermore, Park et al. [140] reported reduced NIV without any damage to the foodstuff with microwave-induced argon cold plasma treatment for 5 seconds. In addition, Hojnik et al. [141] proposed a possibility of degradation of NIV toxicity using argon plasma-generated microwave discharges. Niedźwiedź et al. [142] predicted that the NIV reduction during plasma treatments might be due to structural changes. Besides this, Boeira et al. [143] reported a 56.5% reduction in NIV with magnetic field implicated alcoholic fermentation and proposed this to be an efficient method for NIV degradation.

## 8. Detection Techniques 

NIV and DON generally exist as co-occurring mycotoxins in the food matrices, and thus are detected simultaneously. NIV detection techniques are critical for investigating the incidence, potential prevention, and control of NIV. NIV analysis in foods is critical. However, it is challenging to identify mycotoxins in food because of their matrix complexity. NIV detection techniques are broadly classified as chromatographic and immunological approaches [144].

### 8.1. Chromatographic Techniques

Quantitative identification of mycotoxins is accomplished using chromatographic methods. These methods combine gas or liquid chromatography with mass spectrometry (MS), ultraviolet (UV), or fluorescence (FLD) detectors. These approaches permit the most sensitive detection of mycotoxin [145]. Consequently, reliable analytical methods, such as capillary electrophoresis, and chromatography techniques, such as liquid chromatography (LC) and gas chromatography (GC), ultra-performing liquid chromatography coupled with photodiode array (UPLC-PDA), are reported for the detection of NIV mycotoxins [57,146,147]. Chromatographic methods coupled with UV and FLD detection are primarily used for confirmatory studies, i.e., to confirm or deny the non-compliance with rules previously determined by a screening test. Occasionally, they serve as the validation standard for immunochemically-based diagnostics. In contrast, devices equipped with tandem mass spectrometry (MS/MS) detectors make it possible to adapt the analytical method for NIV detection. The incontestable benefits of MS (including high sensitivity, selectivity, accuracy, and throughput) make it the technique of choice for multi-residue analysis [145]. Due to the intrinsic selectivity of MS/MS detectors, extraction procedures requiring minimal or no sample purification may be effectively established.

Most NIV detection chromatographic methods need a solvent extraction procedure to preferentially transfer the polar organic NIV into a polar solvent combination. Often, methanol and acetonitrile are used for extraction [148]. To increase the accuracy of subsequent detection, solid-phase extraction (SPE) that utilizes selected porous adsorbents and immunoaffinity column extraction that employs antigen–antibody interaction, or a combination of the two approaches, is employed. Immunoaffinity chromatography, known as immunoextraction, is a type of SPE that combines immunological and conventional techniques. These are mainly used as the clean-up before any conventional technique [149]. The QuEChERS (quick, easy, inexpensive, effective, rugged, and safe) sample preparation method utilized in this context greatly simplified analytical procedures and, most importantly, enabled the simultaneous extraction of NIV [150,151,152]. Since typical QuEChERS methods are built on conciliations between optimal extraction conditions for vastly diverse compounds, they are intrinsically wasteful and limit the analytical method’s sensitivity. High-resolution mass spectrometry (HRMS) and tandem mass spectrometry detectors give structural information and potential identification of unknown chemicals. Combining non-selective extraction procedures with mass screening by HRMS or MS/MS enabled the discovery of masked mycotoxins [146]. Instrumental techniques, such as high-performance liquid chromatography (HPLC/UV) [91], thin-layer chromatography (TLC) [153], HPLC [58], and LC-MS/MS [154], are the most often used conventional approaches for the detection of NIV.

HPLC is the most frequently used method for analyzing NIV, depending on the target analytes. It has been coupled with FLD, DAD, and UV detectors. Thus far, HPLC has achieved a low sensitivity and has been utilized in 16 studies to detect NIV. Carballo et al. [155] detected NIV in ready-to-eat food prepared from cereals, legumes, and vegetables using QuEChERS extraction and HPLC-MS/MS. The study resulted in satisfactory extraction, with a 63–119% recovery rate. Ok et al. [91] reported a technique that used an immunoaffinity column for sample cleaning before HPLC-UV detection and quantification. The immunoaffinity clean-up effectively determined NIV and DON simultaneously in rice and bran. The technique was validated before being used to identify analytes in 482 rice and 239 bran samples. Limits of detection and quantification varied from 6.4 to 15.6 and 21.2 to 52.0 mg/kg, highlighting the importance of the immunoaffinity cleaning step in obtaining low limits. Trombete et al. [156] detected the presence of NIV by HPLC-PDA using an immunoaffinity column clean-up coupled with UV in wheat grains for the first time. Gab-Allah et al. [157] investigated the occurrence of NIV, DON, and DON-3-glucoside in wheat and corn flour using ultra-performance liquid chromatography coupled with a photodiode array (UPLC-PDA). Furthermore, Pascale et al. [147] detected NIV using UPLC-PDA in wheat grains.

LCMS/MS has recently become widely employed for detecting NIV without derivatization, providing the most trustworthy findings, while meeting regulatory authorities ([157]. Barthel et al. [66] detected NIV and other trichothecenes in German barley samples (dehulled barley, barley kernels, winter barley, and pearl barley) using LC-MS/MS. Panasiuk et al. [158] investigated NIV detection using LC-MS/MS with different sample preparation techniques. TLC is a low-cost analytical approach for identifying NIV. TLC is based on the interaction between the stationary and mobile phases, allowing for successful separation based on the sample’s molecular structure. A wide range of *Fusarium*-based mycotoxins, including NIV, has been detected using TLC [159]. TLC has been reported to identify the presence of NIV in agricultural products and feed [160]. Bryła et al. [57] reported the occurrence of nivalenol in wheat and reported a 70% chance of NIV occurrence. In addition, De Boevre et al. [161] reported a reliable and simple technique for detecting NIV and its masked form using UHPSFC-MS/MS (ultra-high performance supercritical fluid chromatography coupled with tandem MS).

Unfortunately, these analytical processes need expensive, complex equipment, laborious, time-consuming sample preparation, and a highly qualified technician. Thus, many existing techniques are unsuitable for real-time and on-site implementation, particularly in emergencies. In response to these constraints, various efficient techniques with excellent sensitivity and specificity for detecting and quantifying mycotoxins have been devised to achieve accurate, rapid, sensitive, and cost-effective data [146,149].

### 8.2. Immunological Techniques

Immunoassays help monitor food safety and have been used to analyze NIV [162]. Immunoassays are used for first-level NIV contamination screening because of their simplicity, affordability, sensitivity, and selectivity. The molecular interaction between the target and the biorecognition component, or antibody, is the fundamental principle of immunoassays. As of now, antibodies are undoubtedly the ideal recognition component for immunoassays and biosensors [149]. Kits based on ELISA are commercially available for NIVs and are the most widely utilized analytical method for ensuring food safety across the food chain [163]. Yoshizawa et al. [164] developed an ELISA method for detecting NIV in wheat kernels. Regarding the NIV’s sensitivity, a detection limit of 80 ng/g was recorded. Consequently, ELISA can be utilized as an alternative to conventional detection methods for identifying mycotoxins in cereals and other food matrices.

In addition, immunochemical assays in various formats are continually being developed to provide quick, portable, and user-friendly solutions [145]. The immunochromatographic test (ICT) is the dominant technique. It has frequently been employed for the visual detection and semi-quantitative analysis of NIV [165]. However, in the new scenario of NIV detection, the extreme selectivity of the molecular recognition mechanism, which forbids the simultaneous determination of multiple compounds, the detection of unknown toxins, as well as modified structures created by plant metabolism, appears to pose a potential limitation for the immunochemical-based methods [145].

Some other rapid methods for detecting NIV include immunochromatographic tests, including lateral flow immunoassay [166] and surface plasmon resonance (SPR) immunoassays [167]. However, there are considerable analytical limits, since choosing multiple antibodies with appropriate features is difficult. Due to its distinct advantages, such as quick and accurate target analyte detection, increased sensitivity, and non-destructive method, SPR technology has numerous applications in biosensors and chemical sensors. This method has evolved into a potent analytical method for finding NIV in food and animal feed. SPR also offers details on the affinities, specificities, and kinetic characteristics of biomolecular interactions [168]. To concurrently detect NIV and DON in wheat, an SPR-based competitive inhibition immunoassay was created [168]. In their research, Kadota et al. [167] used a monoclonal antibody that also interacts with NIV and DON. Lateral flow assays are user-friendly, cost-effective, rapid, and extend the food commodity’s shelf life. In this technique, the presence of the targeted toxin is assessed by the depth and length of the color label. Then, the label is visualized using smartphones [149].

### 8.3. Masked Nivalenol as a Significant Concern in Detection

Conjugation of mycotoxin with sugars, which protects plants from toxicity, is known as “masked mycotoxins” and poses a potential threat to food safety [4]. Several studies have shown that masked mycotoxins constitute a possible threat to consumer safety, due to their probable hydrolysis during human digestion, frequently releasing them in their natural form. Masked mycotoxins are mycotoxin derivatives with structural alterations that cannot be detected by traditional detection methods [144]. Nevertheless, the changed form of mycotoxins is typically coupled to proteins or carbohydrates in plants or animals, making their identification challenging. Chemical treatments can remove these modified mycotoxins from plants [110,169]. In general, masked mycotoxins are less hazardous than their parent substances. The most significant problem is investigating the toxicity of modified mycotoxins during their bound phase [170]. Due to the absence of sensitive procedures, identifying masked mycotoxins is challenging. Moreover, the pharmacokinetics of these modified mycotoxins were unclear. Estimating the consumption of these modified mycotoxins, thus, requires a reliable biomarker of exposure. Biomonitoring exposure evaluation is necessary to comprehend the toxicity and mechanism of masked mycotoxins [171]. Nakagawa et al. [172] identified a modified NIV, nivalenol-glucoside (NIVGlc), using LC-Orbitrap HR-MS and tandem mass spectrometry in wheat grains. They reported that about 15% of NIV was modified as NIV-glucosides [172]. In a study, Yoshinari et al. [173] investigated NIV-glucosides derived from NIV-contaminated wheat grain. The masked NIV was identified as NIV-3-O-β-D-glucopyranoside (12–27%) using LC-MS/MS with an immunoaffinity column. Furthermore, Nathanail et al. [150] detected a modified NIV, nivalenol-3-glucoside (NIV-3G), in barley, oats, and wheat using LC-MS/MS. Later, Ksieniewicz-Woźniak et al. [174] detected NIV and NIV-3G in malt and beer using LC-MS/MS coupled with a time-of-flight analyzer. Furthermore, Lee et al. [58] determined NIV and its glucoside derivative (NIV-3-β-D-glucoside) in cereals, legumes and their processed forms using HPLC-UV. Researchers continue working on innovative approaches with enhanced sensitivity, specificity, robustness, timesaving, and cost-efficiency. Rapid procedures are chosen by analysts for rapid detection (e.g., on-site screening of large numbers of samples) or in regular analysis in laboratories where the traditional approach is unavailable. Immunoassays have already found broad applications as screening procedures for mycotoxins, owing to their advantageous characteristics, such as rapidity, simplicity, cost-efficiency, needed sensitivity, and specificity [149].

### 8.4. Degradation Kinetics 

The mitigation of mycotoxin contamination in foods has become paramount, due to the serious damage to the health and economy [175,176,177]. Contamination of NIV can be prevented at different pre- and post-harvest stages; however, this can be degraded at the postharvest stage through various methods, including physical, chemical, and biological methods.

Physical methods include cleaning, sorting, thermal inactivation, microwaves, ultrasound, adsorption, irradiation and others [178]. High temperature treatment or heating is the most basic processing practice applied for softening, taste improvement, degradation of toxic compounds, and sterilization of grains. The heating of NIV in the presence of mild alkaline conditions degraded the NIV into a less toxic compound [139]. However, toxin elimination become tough due to its resistance against various heat treatments and various physicochemical factors [175]. Scudamore et al. [139] reported that the degradation of NIV increased after extrusion cooking, which concluded that the NIV content was not a temperature-dependent compound.

Chemical methods include the use of absorbers, alkalis, and other chemicals to manage different types of trichothecenes. Absorbers are very common for mycotoxin detoxification from food and feed. Activated carbon can bind to several types of mycotoxins, including deoxynivalenol. Mycotoxin contamination can also be minimized by using some bacteria, as they are able to bind to mycotoxin to their cell wall polysaccharides and peptidoglycans [179]. Some lactic acid and propionic acid bacteria have been reported to trap NIV at low pH [180]. Treatment of mycotoxin-containing grains with commonly used chemicals, such as sodium carbonate [181], ammonia and sodium hydroxide [139], sodium bisulfite, sodium metabisulfite and ascorbic acid, converted them into low toxic compounds. Fungicides, such as carbendazim, triazoles, and strobilurins, or combinations of two or more of these compounds, are widely used agrochemically for the management and control of *Fusarium* head blight (FHB). However, dose, timing and host line are responsible for the effectiveness of applied chemicals. As FHB can be caused by various *Fusarium* species, the sensitivity to fungicide can also be a key factor in determining the efficiency of the fungicide group [182,183]. In some cases, when a plant is grown under stressful conditions and in the presence of chemical plant protection products, fungi may produce higher levels of mycotoxins [175].

In addition, biological methods are also being experimented for NIV degradation. Using biological means is found to be much more effective for mycotoxin management. Podgórska-Kryszczuk et al. [175] conducted an experiment to minimize mycotoxin production in wheat grain by *F. culmorum*, *F. graminearum* and *F. poae* by using non-conventional yeast strains (*Candida shehatae* C13, Candida fluviatilis C14, Candida tropicalis C28, *Meyerozyma guilliermondii* K2, *Cyberlindnera saturnus* K10, *Rhodotorula glutinis* E20, *Cryptococcus carnescens* E22). The highest degradation of *F. poae* mycotoxins was recorded with the use of *M. guilliermondii* (K2) and *C. tropicalis* (C28) yeasts, which reduced about 70.4% and 56.3% of NIV in wheat grain during a 14-day incubation test. Moreover, flour obtained from fungal-contaminated grain was used to make model bread. The greatest reduction caused in the NIV content (36.2%) was reported using *M. guilliermondii* (K2). It was reported that the selected yeast isolates were very effective at lowering the content of NIV in bread and these strains were also a better alternative to other chemical food additives [175]. In addition, Boeira et al. [143] reported 56.5% degradation of NIV through alcoholic fermentation with magnetic field application in malted cereals, which altered the profile of the yeast-synthesized oxireductant molecules. During the fermentation process, *Saccharomyces cerevisiae* US-05 synthesizes the glutathione and peroxidase enzyme, which are responsible for the mitigation of NIV. Biological processes, such as oxidation, reduction, hydrolysis, hydration, glycosidation, and isomerization, are also reported for the reduction in the trichothecenes group of mycotoxins [143].

Besides these physical, chemical and biological methods, different plant extracts, such as bioactive compounds, essential oils (EOs), and other phytochemicals have been popularized as green preservatives, as well as known for their antimicrobial properties and widely used for the enhancement of the shelf life of food products [184]. Hope et al. [185] evaluated the antifungal activity of some essential oils and antioxidants. Enhanced NIV production was recorded with the treatment of low concentrations of propyl paraben and cinnamon oil. However, at higher concentrations (500 ppm), butylated hydroxy anisole, cinnamon and clove essential oils were found to be the most effective in the inhibition of NIV production by *F. culmorum*. Treatment of maize seeds with cinnamon, palmarosa, orange, and spearmint essential oils significantly inhibited the growth of *F. graminearum* and *F. culmorum* compared to the control [186]. EOs of rocket, tea tree and rosemary showed significant antifungal activity against ten *Fusarium* spp. isolated from freshly harvested maize in Egypt [187]. Similarly, antifungal activity of cinnamon, oregano, palmarosa, orange, verbena, spearmint, fennel, and rosewood EOs were found to be effective against *F. culmorum* and *F. graminearum* and their secondary metabolites in Polish wheat seeds [188].

### 8.5. Management and Control Strategies

The pathogenic contamination of crops begins at the field level. Timely identification and application of suitable treatment may result in a lower level of a contaminant at the harvesting and storage stage. Mycotoxin contamination can be managed by adoption of good agricultural practices (GAPs), good management practices (GMPs), and good storage practices (GSPs) [8]. These practices mainly focus on reducing the *Fusarium* spp. contamination and minimize the production of mycotoxins in crops in the field by the adoption of cultural practices, such as crop rotation, ploughing, fertilizer application, planting date and harvest times, genetics approaches, such as host resistance, inclusion of chemicals, e.g., fungicide and/or biocontrol application, and development of forecasting models [182].

Crop rotation has a significant effect on the determination of disease level of the crops in fields. Practicing crop rotation significantly reduces primary inoculum that is present during cropping season. Deep tillage is more beneficial compared to minimum or no tillage practices [182]. Minimum or no tillage indorses the development of *Fusarium* spp. because plant residue left on the ground can infect the subsequent crop. Removal of plant residue and clean cultivation are also beneficial to minimize fungal contamination. The date of planting often determines the date of flowering and environmental conditions at flowering, which are very crucial stages for of FHB disease incidence. This is adopted to set up the flowering time when weather conditions are not suitable for the onset of disease practices [182]. Selection of variety is one of the major factors in disease management. The variation in the level of susceptibility or resistance of particular cultivars against *Fusarium* spp. infection is due to differences in agronomic practices, environmental conditions, and genetic pools of breeding programs in different countries [189]. Multiple breeding programs are involved in the development of resistant varieties and newly developed varieties are also found to be less impacted compared to sensitive varieties. Development and study of neural models are found to be useful to analyze the concentration of mycotoxins in winter wheat grain [190]. Cropping systems and crop cycles also affect contamination by *Fusarium* spp. The dominance of *Fusarium asiaticum* was reported in wheat crop cultivated in a wheat-rice rotation, compared to the dominance of *F. graminearum* in wheat-maize rotations [182]. Multi location studies in China and Japan have reported that the location of the crop also affects the severity of fungal species and toxin produced [191,192]. In Poland, higher FHB incidence was reported in wheat when it was planted after wheat or corn, as compared to sugar beet in pre-crop [193].

Postharvest losses of food material due to fungal/ microbial contaminations are considered to be worse compared to insect or rodent damage. The first step of good postharvest quality management starts with harvesting. A healthy crop should be harvested at the proper time of maturity, with a proper harvesting method to reduce mechanical damage. The transport vehicle and containers should be free from any contaminants. After harvesting, proper shorting, grading and drying can improve the storability and reduces contaminants [194]. *Fusarium*-infected grains are generally shriveled, small, and lighter than healthy grains, so infected and non-infected grains can be distinguished by their physical characteristics. Sorting and cleaning process removes the infected small seeds, as well as dust, dirt and foreign materials [182]. A proper drying method, storage at optimum moisture content, improved storage infrastructure, proper aeration and pest management are generally able to maintain good quality of grains during storage, which maintain postharvest quality and reduces postharvest loss. There are multiple storage conditions that are recommended to reduce contamination; however, the efficacy varies according to their geography and climatic condition [194]. Adoption of proper storage conditions and maintaining optimum moisture levels play a significant role in fungal and mycotoxin development. Storage temperature and lower water activity has shown to reduce accumulation of NIV in maize produced by *Fusarium meridionale* and *F. boothii* [195]. Proper storage conditions of Korean ginger were also found to be effective in the reduction in NIV where samples were stored at 13℃ and 96% RH, which significantly reduced the NIV content in ginger [196]. Prevention from the fungal disease and mycotoxin production is more effective than management and removal of mycotoxin. So, adoption of GAPs, GSPs, GMPs and incorporation of the Hazard Analysis Critical Control Point (HACCP) in postharvest management strategy would be more effective in managing NIV contamination [112]. Furthermore, awareness initiatives, such as teaching and training at the level of farmers, may also be very beneficial to prevent the contamination and production of mycotoxin at each step involved, from the farm to the fork.

## 9. Conclusions

Consumption of nivalenol (NIV)-contaminated foods by humans and animals worldwide carries serious health concerns. The toxin could enter food chains by contaminating foods at any stage of the pre- and post-harvest conditions. The cytotoxicity of NIV increases the risk to human and animal health. Adequate hazard analysis critical control point techniques, good agricultural practices, and good manufacturing practice may be required to successfully regulate the toxins during various agricultural and processing stages. Other physical, chemical, and biological strategies could be employed to lessen NIV formation and contamination, in addition to the usage of essential oils and phytochemicals. Additionally, the toxin is reported to exist in masked forms. Hence, in order to identify, quantify, and mitigate them, quick and accurate detection procedures become crucial, as well as difficult. This emphasizes the need for clear and dependable detection techniques for their control. A concealed concern for food safety and security exists, since there is little information available on the masked forms of NIV in products and because these forms are likely to go unnoticed and underreported. Hence, future research should focus on a thorough examination of the masked forms of NIV.

## Figures and Tables

**Figure 1 toxins-14-00527-f001:**
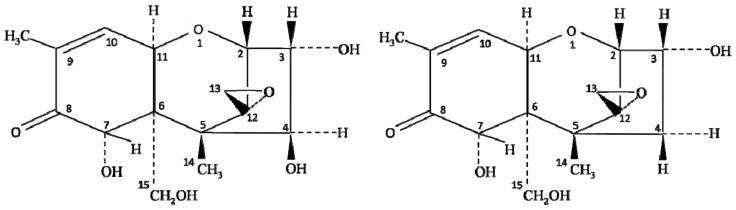
Chemical structure of nivalenol (NIV) and deoxynivalenol (DON).

**Figure 2 toxins-14-00527-f002:**
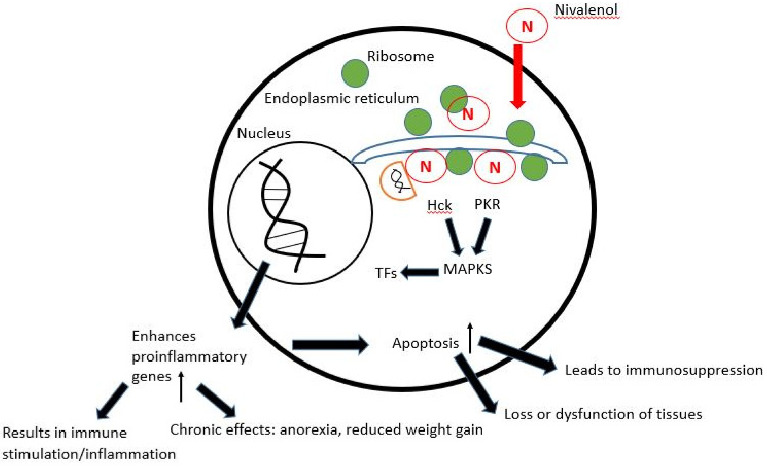
Proposed mechanism of action of NIV.

**Table 1 toxins-14-00527-t001:** Occurrence of nivalenol in food and feed around the world.

Food Matrix	Country	Range (μg/kg)	Detection Technique	References
Food				
Adlay millet	South Korea	12.6–337.6	HPLC-UV	[58]
Baby foods	Spain	75–100	HPLC-MS/MS	[59]
Baby formula	South Korea	4.4–1000	HPLC-UV	[60]
Barley	England	10–1088	GC/MS	[61]
	South Korea	10.4–110.3	HPLC-UV	[58]
	Italy	21.7–106	LC-MS/MS	[62]
	Spain	12.47	GC-MS	[63]
Baked snacks	Spain	55.7	GC-MS	[64]
Barley grain	Poland	5	TLC and HPLC	[65]
Barley and barley products	Germany	0.87–19	LC-MS/MS	[66]
Beer	Czech Republic	4–6	UHPLC-APCI-Orbitrap MS	[67]
Beer	Spain	10–15	UHPLC-APCI-Orbitrap MS	[68]
Black bean paste (Chunjang)	South Korea	83.8	HPLC-UV	[58]
Breakfast cereals	Spain	51.1–106.5	GC-MS	[64]
Breakfast cereals	South Korea	1096.8	GC-MS	[69]
Brown rice	South Korea	47.4	HPLC-UV	[58]
Cereals	Finland	185–300	LC-MS/MS	[70]
Cereals	Czech Republic	50	UHPLC-ESI-ToF-MS	[71]
Cereal based products	Czech Republic	25	UHPLC-APCI-MS/MS	[72]
Cereal based products	Switzerland	100	HPLC-ESI-MS/MS	[73]
Cereals and cereal-based products	Spain	121–176	LC-MS/MS	[74]
Corn	South Korea	0–181.41 *	HPLC-PAD	[75]
Corn	France	7–340	HPLC	[76]
De-hulled and naked barley	Spain	1.1–7.6	LC-MS/MS	[66]
Durum wheat	France	60	HPLC	[76]
Durum wheat flour	Denmark	83–440	GC-ECD	[77]
Flour bread	Italy	5–8	LC-MS/MS	[62]
Foxtail millet	South Korea	27.4–370.8	HPLC-UV	[58]
Ground wheat	Italy	3.5–63.5	LC-APCI-MS/MS	[78]
Groundnut-maize based snacks	Nigeria	1.8–2.5	LC-MS/MS	[79]
Grain-based product	Italy	30	GC-MS	[80]
Groundnut	Nigeria	1.0	LC-MS/MS	[79]
Malting barley	Spain	35	LC-MS/MS	[66]
Maize	South Korea	51.3	HPLC-UV	[58]
	Nigeria	0.8	LC-MS/MS	[79]
	China	2.1–15.3	UHPLC-MS/MS	[81]
	Austria	22.3–250	LC-MS/MS	[82]
	Spain	6.4	GC-MS/MS	[83]
	Germany	4.41–20	GC	[84]
	UK	5–10	HPLC	[85]
	Poland	2	TLC and HPLC	[65]
Maize flour	Germany	39	GC-MS	[86]
Maize-based breakfast cereal	Spain	16–60.2	GC-MS	[64]
Multicereal Flour	Spain	75	LC-MS/MS	[87]
Mixed paste	South Korea	15.9–100.6	HPLC-UV	[58]
Mixed grains	South Korea	88.9	GC-MS	[69]
Oats	South Korea	23.5	HPLC-UV	[58]
	Italy	45.5–50.4	LC-MS/MS	[62]
	Germany	17	LC-MS/MS	[88]
	Italy	8–20	LC-MS/MS	[89]
	Austria	100	HPLC	[76]
	England	10–112	GC/MS	[90]
Oats grain	Poland	6	TLC and HPLC	[65]
Pearl barley	Spain	0.18	LC-MS/MS	[66]
Popcorn	South Korea	68.7	HPLC-UV	[58]
Red chili paste (Gochujang)	South Korea	8.5–120.2	HPLC-UV	[58]
Rice	South Korea	10	HPLC	[91]
Rice	Thailand	0.50–15.00	UHPLC-MS/MS	[92]
Rice wine	South Korea	2.5	HPLC-UV	[60]
Rye	Italy	33.9–34.4	LC-MS/MS	[62]
	Germany	1.8	LC-MS/MS	[88]
	France	2–48	HPLC	[76]
Rye flour	Denmark	38–48	GC-ECD	[77]
Rye grain	Poland	5	TLC and HPLC	[65]
Sesame butter	China	0.05–7.25	UHPLC-MS/MS	[93]
Semolina	Germany	36	GC-MS	[86]
Wheat	Italy	12–106	LC-MS/MS	[62]
	Japan	0.2	HPLC-AAPI-MS/MS	[94]
	Spain	53.6	GC-MS/MS	[83]
	Germany	33	GC-MS	[95]
	Poland	10	GC-GC-ToF-MS	[96]
	Argentina	0.11–0.40	HPLC	[97]
	England	10–330	GC/MS	[98]
Wheat flour	South Korea	31.8	GC-MS	[69]
	Denmark	10–189	GC-ECD	[77]
	Spain	30	HPLC-ESI-MS/MS	[99]
Wheat semolina	Spain	8.8–13.6	GC-MS/MS	[100]
Winter barley	Spain	5.6–6.5	LC-MS/MS	[66]
Winter wheat	Italy	70	HPLC-MS/MS	[101]
Spelt	Italy	23	LC-MS/MS	[62]
**Feed**				
Bran	South Korea	11.1–36.9	HPLC-UV	[91]
Cattle feed	South Korea	0–111.52 *	HPLC-PAD	[75]
Chicken feed	South Korea	0–101.23 *	HPLC-PAD	[75]
Maize silages	Denmark	122	LC-MS/MS	[102]
Pig feed	South Korea	0–84.21 *	HPLC-PAD	[75]
Wheat germ	Germany	26	GC-MS	[86]
Wheat bran	Germany	37	GC-MS	[86]

* ng/kg of samples. GC: gas chromatography; GC-MS: gas chromatography–mass spectrometry; GC-ECD: gas chromatography with electron capture detection; GC-GC-ToF-MS: two-dimensional gas chromatography/time-of-flight-mass spectrometry; GC-MS/MS: gas chromatography coupled with tandem mass spectrometry; HPLC-AAPI-MS/MS: high performance liquid chromatography/atmospheric pressure ionization/tandem mass spectrometry; HPLC-APCI-MS/MS: high-performance liquid chromatography–atmospheric pressure chemical ionization–tandem mass spectrometry; HPLC-ESI-MS/MS: high-performance liquid chromatography/electrospray ionization tandem mass spectrometry; HPLC-MS/MS: high performance liquid chromatography and tandem mass spectrometry; HPLC-PAD: high-performance liquid chromatography-photodiode array detector; HPLC-UV: high performance liquid chromatographic method coupled with ultraviolet detector; LC-APCI-MS/MS: liquid chromatography coupled with atmospheric pressure chemical ionization triple quadrupole mass spectrometry; LC-MS/MS: liquid chromatography–tandem mass spectrometry; TLC: thin layer chromatography; HPLC: high performance liquid chromatography; UFLC-MS/MS: ultra-fast liquid chromatography tandem mass spectrometry; UHPLC-APCI-MS/MS: ultra-performance liquid chromatography–atmospheric pressure chemical ionization tandem mass spectrometry; UHPLC-APCI-Orbitrap MS: ultrahigh-performance liquid chromatography electrospray ionization quadrupole Orbitrap high-resolution mass spectrometry; UHPLC-ESI-ToF-MS: ultra-high-performance liquid chromatography coupled with electrospray ionization quadrupole time-of-flight mass spectrometry; UHPLC-MS/MS: ultra-high performance liquid chromatography tandem mass spectrometry; UHPLC-QqLIT-MS: ultra-high-performance liquid chromatography quadrupole linear ion trap mass spectrometry.

## Data Availability

The data presented in this study are available in this.

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
