# Peer review of "Nivalenol Mycotoxin Concerns in Foods: An Overview on Occurrence, Impact on Human and Animal Health and Its Detection and Management Strategies"

_toxins, 2022, doi:10.3390/toxins14080527_

Round 1
Reviewer 1 Report
I carefully read the manuscript entitled “Nivalenol Mycotoxin Concerns in Foods: An Overview on Occurrence, Impact on Human and Animal Health and its Detection and Management Strategies”. The authors have well developed the various points and described them in a very exhaustive way. In addition, the topic is of interest and is in line with the journal’s aim. I recommended the publication of the present review, after checking some minor typos/mistakes.
I suggest a check of the English, especially with regard to the typos or the use of plural/singular form: e.g. in the abstract ‘fungi’ instead of ‘fungus’; title of paragraph 2 ‘major sources’ instead of ‘major source’; check the dot (page 5, line 101) after the abbreviations reported in the legend of Table 2;
I suggest to rephrase the first sentence of introduction;
Legend of Table 2: find a way to avoid to repeat the meaning of the abbreviation more times in order to shorten it. Moreover, ‘ToF’ instead of ‘TOF’; e.g. ‘MALDI-ToF’;
Try to enhance Figures resolution. Moreover, try to improve the content of Figure 3;
Check reference number 75. It is not in line with the journal’s guidelines.
Author Response
PFA

Reviewer 2 Report
This review is very well written and organized.
For the reviewer with experience in the research it was very interesting to read this article. As the researchers working with the chromatographic methods for the analysis of multi-class mycotoxins of Fusarium varieties, NIV is a common companion of DON and other Type B trichothecenes. The results of several recent studies have indicated even higher or comparable levels of NIV compared to that of DON raising concerns of this mycotoxin.
Then Introduction part is brief and maybe formally it should be corrected by adding the information of the main aspects of providing this overview. It is already mentioned in the abstract, but maybe some notes , taking into account several recently published studies on reviewing DON, NIV and other Type B trichothecenes in cereals and their products.
The tittle of the section 2 maybe should be rechecked as it indicate not "major source", but the main used instrumental methods used for mycotoxin analysis (the reviewer did not check them all, but it would be interesting to mention how many of them were single methods or most of them were used for multi-mycotoxin analysis).
It would be also maybe recommended to provide some discussion of the results of Table 1.
It is not clear in what order the results are grouped as there are rather different matrices grouped. Msaybe it could be interesting to show table for nonprocessed cerals and processed ones, including slod matices and beer. There are different types and factors.
As well, many of the methods are still not so commonly used, such as GC-MS methods, compared to HPLC-UV and HPLC-MS methods with different detection modes and MS resolution capacities.
The limitations for mycotoxins in feed are different, maybe there should be more attention related to non processed and processed cereals.
It seems that random seasons, origins and procession factors may have impact on the results. Maybe it should be also discussed more deeply within the presentation of this table.
However, maybe it would be recommended to start after Introduction with the section of NIV chemistry and biosynthesis , followed by the Environmental factors on the NIV production (fungi species, microclimate conditions).
Thus the section 2. and Table 1 seems to fit better after these sections within the discussion of the methods of analysis.
But this is just a suggestion and is up to the authors.
Some suggestions of minor changes may include:
- Line 6: Fusarium in Italics (the same Line 68 and check elsewhere in the text)
- Line 64: overview the tittle of the Section 2
- Line:74: Īn various food and worldwide"- rephrase as it is more attributed to cereal (e.g., grain) based products and non-processed cereals.
Maybe the section of masked NIV sholud be replaced as a seperate section with also small table indicating the occurence data.
Also it would be recommended to place the mitigation methods including the reduction in processed foods after these sections with added also information on the other methods of treatment. There should be also a seperate discussion for non-processed cereals and the cereal based foodstuffs.
These ae also recommendations.
In general this is an excelent review and it should be published.
It has also important discussion for the exposure evaluation based on the recent data and techniques that will be very improtant for the researchers wanting to evaluate their data with some calculation aproaches in order to estimate the occurence rate and effects on the consumers of NIV and DON rich products.
Author Response
PFA
